# PFGS: POSE-FUSED 3D GAUSSIAN SPLATTING FOR COMPLETE MULTI-POSE OBJECT RECONSTRUCTION

## ABSTRACT

Recent advances in 3D Gaussian Splatting (3DGS) have enabled high-quality, real-time novel-view synthesis from multi-view images. However, most existing methods assume the object is captured in a single, static pose, resulting in incomplete reconstructions that miss occluded or self-occluded regions. We introduce PFGS, a pose-aware 3DGS framework that addresses the practical challenge of reconstructing complete objects from multi-pose image captures. Given images of an object in one main pose and several auxiliary poses, PFGS iteratively fuses each auxiliary set into a unified 3DGS representation of the main pose. Our pose-aware fusion strategy combines global and local registration to merge views effectively and refine the 3DGS model. While recent advances in 3D foundation models have improved registration robustness and efficiency, they remain limited by high memory demands and suboptimal accuracy. PFGS overcomes these challenges by incorporating them more intelligently into the registration process: it leverages background features for per-pose camera pose estimation and employs foundation models for cross-pose registration. This design captures the best of both approaches while resolving background inconsistency issues. Experimental results demonstrate that PFGS consistently outperforms strong baselines in both qualitative and quantitative evaluations, producing more complete reconstructions and higher-fidelity 3DGS models.

## 1 INTRODUCTION

High-fidelity 3D object reconstruction from multi-view images is fundamental to modern computer vision and graphics, powering applications in immersive content creation, virtual and augmented reality, robotics, and digital twins. Among recent advances, 3D Gaussian Splatting (3DGS) (Kerbl et al., 2023) excels at real-time novel-view synthesis, providing a compact, continuous, and photorealistic representation of radiance fields. To obtain high-quality 3DGS reconstructions, current methods assume that the target object remains centered and stationary throughout capture.

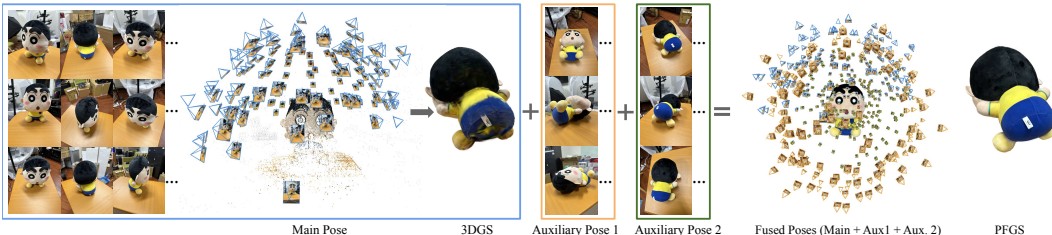

Main Pose    3DGS   Auxiliary Pose 1 Auxiliary Pose 2  Fused Poses (Main + Aux1 + Aux. 2)   PFGS

Figure 1: **PFGS: Pose-Fused Gaussian Splatting for Complete Multi-Pose 3D Reconstruction.** Starting from a main-pose capture and an auxiliary-pose sequence, our method fuses both into a unified 3D Gaussian Splatting model. PFGS performs robust multi-pose alignment and refinement to integrate diverse viewpoints, completing geometry and appearance across self-occluded and hard-to-see regions. The result is a photorealistic and view-consistent reconstruction that surpasses single-pose limitations.

Occlusion, support constraints, and limited physical access often necessitate changing the object's pose during image acquisition. Specifically, a user first captures a set of multi-view images in a *main pose* and then acquires additional images in one or more *auxiliary poses* to form *multi-pose image captures*, enabling comprehensive object reconstruction. This workflow-common in turntable-based or handheld scanning-enables the recovery of occluded or hidden surfaces without redundant imaging. Although conceptually straightforward, multi-pose reconstruction introduces several technical challenges. First, the object's pose changes across capture sessions, making it impossible to jointly estimate camera parameters with standard Structure-from-Motion (SfM) techniques (Schönberger & Frahm, 2016), which expect a static object relative to the background. Applying SfM naively to the combined image set often results in mismatched or erroneous camera poses. Second, cross-pose variations in appearance and geometry undermine reliable correspondence estimation and alignment, particularly when relying solely on foreground cues. Recent advances in multi-view 3D foundation models—such as MapAnything (Keetha et al., 2025), VGGT (Wang et al., 2025), Fast3R (Yang et al., 2025) and MASt3R (Leroy et al., 2024)—have demonstrated improved robustness over traditional SfM. However, our experiments show that these models struggle with large-scale datasets (hundreds of views), yielding noisy or prohibitively expensive results even on high-end GPUs like the NVIDIA RTX 4090. Third, merging independently reconstructed 3DGS models without careful correction can introduce visual artifacts such as ghosting, splat duplication, and radiance inconsistency.

To address the multi-pose challenge, we introduce **P**ose-**F**used 3D **G**aussian **S**platting (PFGS), a pose-aware framework for complete 3DGS object reconstruction, as illustrated in Figure 1. PFGS reconstructs a multi-pose object by fusing image sets captured in a main pose and an auxiliary pose through a three-stage pipeline each time: *global registration*, *local registration*, and *3DGS model completion*. We first estimate initial camera poses and sparse points for each set and segment the foreground regions for the main-pose and auxiliary-pose, respetively; these outputs are used to build an initial 3DGS from the main-pose data. Global registration aligns the auxiliary views to the main-pose coordinate system using a *mix-pose image set*. We align the main- and auxiliary-pose views guiding by the estimated mix-pose views as via a *silhouette-consensus fusion* strategy that robustly registers all auxiliary cameras to the main-pose coordinate frame. Local registration further refines these alignments of auxiliary cameras. Finally, the aligned multi-pose data is used to fine-tune the 3DGS representation, yielding a unified and complete reconstruction. By iteratively applying these stages, PFGS can incorporate additional auxiliary poses and progressively improve the completeness of the reconstruction. To facilitate a rigorous pose evaluation, we construct a synthetic dataset comprising five diverse objects, designed to mimic in-the-wild capture scenarios. In addition, to assess the effectiveness of our method under real-world conditions, we collect a complementary dataset consisting of five physical objects.

Experimental results demonstrate that PFGS consistently outperforms baseline methods in both qualitative fidelity and quantitative metrics, producing more complete reconstructions and superior novel-view synthesis.

Our main contributions are summarized as follow.

- We present PFGS, a practical and efficient framework for complete object reconstruction from multi-pose image captures incrementally for 3D Gaussian Splatting.

- We propose an effective global registration method for aligning image sets captured with the object placed in different poses.

- We demonstrate that PFGS outperforms baselines in both camera registration and 3DGS reconstruction quality on synthetic dataset, and is robust on real-world multi-pose captures.

## 2 RELATED WORK

### 2.1 MULTI-VIEW RECONSTRUCTION AND NOVEL VIEW SYNTHESIS

Traditional multi-view pipelines reconstruct 3D geometry of the scene from overlapping images taken under a single pose configuration of the objects. Structure-from-Motion (SfM) first solves pixel correspondences and estimates intrinsic and extrinsic camera parameters (Özyeşil et al., 2017). Widely used systems such as ORB-SLAM (Mur-Artal et al., 2015) and COLMAP (Schönberger

& Frahm, 2016) combine feature extraction, matching, triangulation, and reconstruction. Co-SLAM (Wang et al., 2023a) further improves accuracy by jointly optimizing features, matches, and camera poses, while VGGSfM (Wang et al., 2023b) introduces an end-to-end differentiable SfM pipeline for fully integrated optimization. Nevertheless, the result of SfM still relies on the sequential structure of the input images, making it vulnerable to sparse or multi-pose image captures.

Recent pointmap methods replace traditional SfM by predicting 3D points per image and aligning them to predict camera poses. DUSt3R (Wang et al., 2024) uses pairwise transformers but is limited to image pairs, while MASt3R (Leroy et al., 2024) scales to all views with a memory mechanism. Spann3R (Wang & Agapito, 2024) adds spatial memory for incremental fusion across wide baselines, and Fast3R (Yang et al., 2025) employs a lightweight two-stage decoder for real-time reconstruction of thousands of images. VGGT (Wang et al., 2025) unifies camera estimation, depth prediction, and pointmap generation in a single feed-forward transformer. Therefore, pointmap methods alleviate the sequential constraints of SfM and improve robustness to challenging captures.

Novel view synthesis, grounded in multi-view reconstruction, aims to generate photorealistic renderings of a 3D scene from previously unseen viewpoints. Neural Radiance Fields (NeRF) (Yang et al., 2021) represent a continuous 5D radiance field using a multilayer perceptron (MLP), where volumetric ray marching integrates color along camera rays. Although NeRF achieves high visual fidelity, it is computationally expensive, with long training times and high inference latency caused by dense sampling and repeated neural evaluations. In contrast, 3D Gaussian Splatting (3DGS) (Kerbl et al., 2023) employs an explicit set of Gaussian primitives, enabling real-time rendering and training that typically converges within minutes. The quality of novel view synthesis is often bottlenecked by multi-view reconstruction, which breaks down under pose misalignments. Multi-pose captures exacerbate this issue, as background variations confuse both SfM and pointmap-based methods. We address this challenge with a pose-aware fusion strategy that explicitly aligns and merges camera poses for robust 3DGS reconstruction.

## 2.2 ONLINE 3DGS RECONSTRUCTION

Reconstructing dynamic objects via online streaming poses unique challenges. To achieve this, CF-3DGS (Fu et al., 2024) proposes an SfM-free pipeline that processes input images sequentially, incrementally growing Gaussians and updating camera poses as each new frame arrives. StreamGS (Li et al., 2025) extends this approach to unposed image streams by predicting and fusing per-frame Gaussians on the fly, while GSFusion (Wei & Leutenegger, 2024) integrates TSDF fusion with Gaussian splatting to merge RGB-D frames into a unified map and prune redundant splats for memory efficiency. However, like classical SfM, these methods assume a sequential input structure, making them unsuitable for multi-pose image captures.

## 2.3 OBJECT-FOCUSED 3DGS RECONSTRUCTION AND RESTORATION

Object-focused pipelines such as ObjectNeRF (Yang et al., 2021) and Co3D (Reizenstein et al., 2021) exploit category-level priors or multi-instance fusion to streamline content creation. Recent 3D Gaussian Splatting based approaches, including HOGSA (Qu et al., 2025), BIGS (On et al., 2025), and GaussianObject (Yang et al., 2024), further advance object-centric capture for bimanual hand-object interaction understanding and sparse-view reconstruction. However, these methods rarely address the challenge of consolidating several partial reconstructions of the same object acquired under different poses into a single, watertight geometry, which is critical for reliable downstream use.

A complementary avenue is to synthesize or restore intermediate views that bridge disparate poses and provide additional supervision. RI3D (Paliwal et al., 2025) and Difix3D+ (Wu et al., 2025) follow this idea by using diffusion models (Ho et al., 2020; Rombach et al., 2022) to suppress artifacts and recover fine details. However, for multi-pose object reconstructions, these restoration techniques would struggle to enforce cross-pose consistency through the diffusion prior.

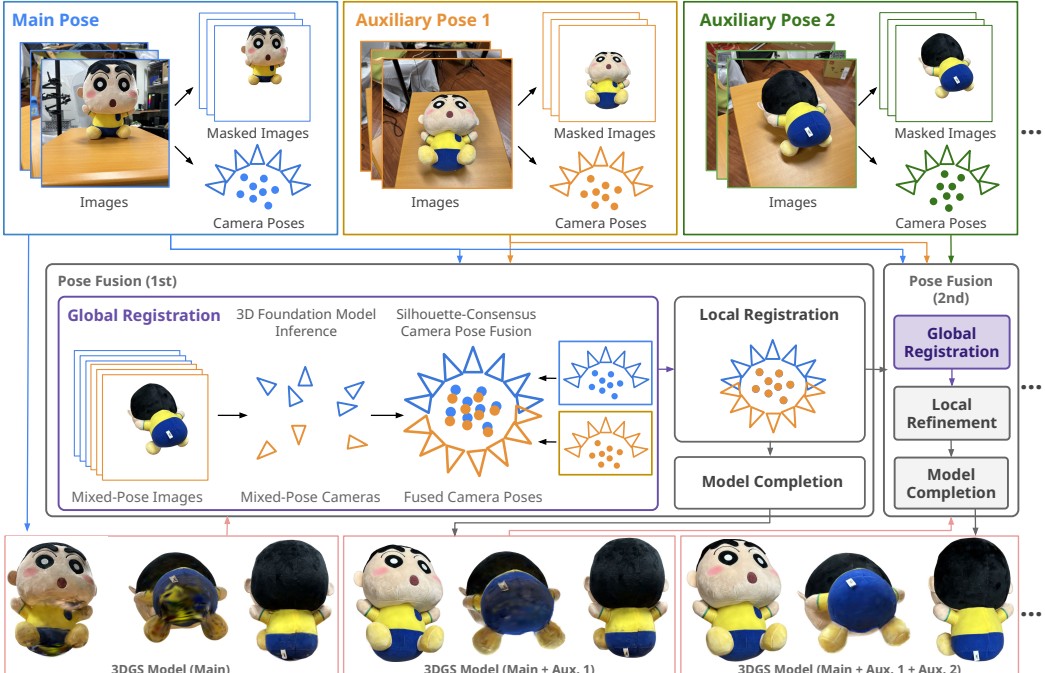

Figure 2: **System overview of PFGS.** Top row: multi-view image sets are captured in a main pose and one or more auxiliary poses; each set is preprocessed with background segmentation and SfM to estimate initial camera poses and sparse geometry. Middle row: we fuse the main pose with the first auxiliary pose via global registration, local registration refinement, and 3DGS completion. Within global registration, we select mixed-pose images from both sets and estimate their cameras using a multi-view 3D foundation model; a silhouette-consensus camera-pose fusion then registers the auxiliary-pose cameras to the main-pose cameras using the mixed-pose cameras as reference. Bottom row: the 3DGS is updated incrementally by repeating this fusion for the first and subsequent auxiliary poses.

## 3 METHODOLOGY

To obtain a complete 3DGS model of an object, users capture a main-pose image set and then acquire additional views from one or more auxiliary poses to expose occluded geometry. To handle pose changes and reliably merge all views despite SfM failures and cross-pose variations, we propose PFGS. Figure 2 depicts the PFGS pipeline. Given a main-pose and an auxiliary-pose image set, we first build a 3DGS from the main-pose images. We then estimate camera poses with COLMAP and extract object masks with SAM2 for both sets, respectively. The key component of each fusion of the paired datasets is the global registration via a three-step process:

(i) we select a subset of visually overlapping views across poses (*mixed-pose images*);

(ii) we estimate camera poses of the mixed-pose images using a multi-view 3D foundation model (*mixed-pose views*), and

(iii) we alignment the main-pose and auxiliary-pose views with the mixed-pose view through a *silhouette-consensus fusion* strategy, which robustly registers all auxiliary cameras to the main-pose coordinate frame.

This multi-step design balances efficiency and robustness. It avoids the instability of applying foundation models to large image sets, while also addressing the limitations of traditional SfM in multi-pose scenarios. Next, we perform local registration to further refine the auxiliary camera poses. Finally, we conduct 3DGS model completion using all aligned views to fine-tune the 3DGS model. For additional auxiliary poses, we repeat this routine, using the fused camera poses and fused 3DGS as the main part and incorporating one auxiliary pose incrementally.

## 3.1 GLOBAL REGISTRATION

The goal of the global registration stage is to align the auxiliary-pose cameras to the main-pose coordinate system, enabling consistent multi-pose reconstruction. This is challenging due to cross-pose appearance changes and the inability of traditional SfM to operate across non-rigid image sets. To address these challenges, PFGS adopts a multi-step registration pipeline that combines feature-based image selection, 3D foundation model inference, and silhouette-consensus pose alignment.

### 3.1.1 MIXED-POSE IMAGE SELECTION

We begin by selecting a subset of representative images across the main and auxiliary poses, which we refer to as mixed-pose images. These serve as input to the 3D foundation model for pose estimation. To select $M$ images from the main set and $N$ images from the auxiliary set, we first encode all candidate images using the DINOv2 (Oquab et al., 2023) image encoder to obtain feature descriptors. Cosine similarity is then computed between all main–auxiliary image pairs. We retain the top-ranked $K$ pairs based on similarity and subsequently perform geometric verification using VGGT. Specifically, VGGT predicts camera poses for each candidate pair, and we discard pairs if the angle between their forward vectors exceeds $\phi$ degrees or if the angle between their up vectors exceeds $\delta$ degrees. This step enforces geometric consistency, which is crucial since multi-view foundation models often fail when input images exhibit significantly divergent viewing directions. For each verified top-$K$ pair, we expand the selection by identifying the $M - 1$ nearest neighbors of the main image in the main set, measured in its own camera coordinate space, and the $N - 1$ nearest neighbors of the auxiliary image in the auxiliary set, measured in its corresponding camera coordinate space. We then compute the mean similarity across all pairwise combinations of the resulting $M$ and $N$ images. This procedure is repeated for all verified top-$K$ pairs, and the expanded set corresponding to the highest mean similarity score is selected as the final representative $M$ and $N$ images. In this way, the selected mixed-pose images are both geometrically consistent and well-aligned in terms of viewing direction, thereby providing reliable input for subsequent pose estimation.

### 3.1.2 3D FOUNDATION MODEL INFERENCE

The selected mixed-pose images are fed into Fast3R (Yang et al., 2025) in a single forward pass, yielding a set of jointly predicted camera poses in a shared coordinate frame. However, these poses are in an arbitrary frame and need to be aligned to the original COLMAP reconstruction of the main pose.

### 3.1.3 SILHOUETTE-CONSENSUS POSE FUSION

After estimating camera poses for the mixed-pose images using Fast3R, we perform a two-stage *silhouette-consensus fusion*, $\mathcal{F}$, to align all cameras into the coordinate frame of the main-pose COLMAP reconstruction. This process estimates optimal similarity transformations (rigid and scale) that minimize the silhouette misalignment between rendered 3DGS masks and foreground masks produced by SAM2.

We begin by defining a unit function,

$$(T, s) = \mathcal{F}(P_{\text{src}}, P_{\text{tgt}}, P_{\text{ref}}), \tag{1}$$

which estimates the rigid transformation $T$ and scale factor $s$ that align a source pose set $P_{\text{src}}$ to a target pose set $P_{\text{tgt}}$, evaluated by silhouette consistency over a reference set $P_{\text{ref}}$. The function's procedure is detailed in Algorithm 1.

To unify all camera poses within the main-pose coordinate frame, we apply the silhouette consensus fusion in two stages. Each stage aligns one subset of poses using a silhouette-guided similarity transformation (see Figure 3).

**Stage 1: Align the mixed-pose coordinate frame with the main-pose coordinate frame (Figure 3(left))** We first align the predicted mixed-pose Fast3R cameras with the main-pose COLMAP reconstruction. We estimate the optimal transformation by solving

$$(T^*, s^*) = \mathcal{F}(P_{\text{mix}}^{\text{main}}, P_{\text{main}}, P_{\text{mix}}^{\text{aux}}), \tag{2}$$

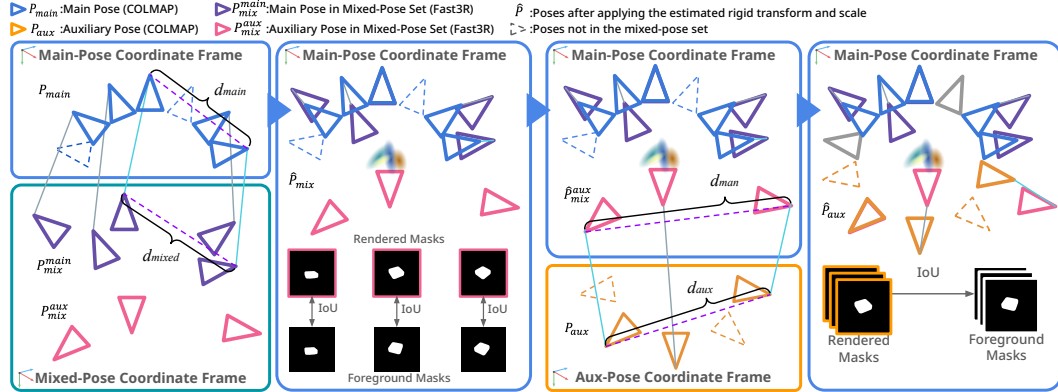

Stage 1: Align Main Poses          Stage 2: Align Auxiliary Poses

Figure 3: **Two-stage silhouette-consensus pose fusion.** We align all cameras to the main-pose coordinate frame through two stages. *Stage 1*: Align the mixed-pose coordinate frame with the main-pose coordinate frame. We select a pair of main-pose cameras from the mixed-pose Fast3R predictions ($P_{mix}^{main}$) and their corresponding cameras from the COLMAP reconstruction ($P_{main}$). A rigid transformation and scale ($d_{main}/d_{mixed}$) are computed from the selected pairs and applied to all Fast3R predictions. The alignment is evaluated by rendering main-pose 3DGS masks using the transformed auxiliary cameras in the mixed-pose group, and computing the average IoU against the corresponding SAM2 masks. The transformation yielding the lowest average IoU is selected. *Stage 2*: Align the auxiliary-pose coordinate frame with the transformed mixed-pose coordinate frame (now aligned to the main-pose frame) using the same process. This produces a unified multi-pose camera configuration in the main-pose coordinate system for downstream optimization.

---

**Algorithm 1** Silhouette-consensus fusion, $\mathcal{F}(P_{\text{src}}, P_{\text{tgt}}, P_{\text{ref}})$

---

1: best_IoU $\leftarrow 0$
2: **for all** pairs $(p_{s1}, p_{s2}) \in P_{\text{src}}$ **do**
3:     Find corresponding $(p_{t1}, p_{t2}) \in P_{\text{tgt}}$
4:     $T$: Align $p_{s1} \rightarrow p_{t1}$ by matching position and forward direction
5:     Estimate scale factor:
$$s = \frac{\|p_{t1} - p_{t2}\|}{\|p_{s1} - p_{s2}\|}$$
6:     $P'_{\text{ref}}$: Apply $(T, s)$ to all poses in $P_{\text{ref}}$
7:     Render main-pose 3DGS masks using $P'_{\text{ref}}$; compute average IoU with corresponding SAM2 masks
8:     **if** average IoU > best_IoU **then**
9:         best_IoU $\leftarrow$ average IoU; store $(T, s)$
10:     **end if**
11:     Repeat the above steps by aligning $p_{s2} \rightarrow p_{t2}$
12: **end for**
13: **return** optimal $(T, s)$

---

where $P_{\text{mix}}^{\text{main}}$ and $P_{\text{mix}}^{\text{aux}}$ represents main-pose cameras and auxiliary-pose cameras estimated from mixed-pose images and $P_{\text{main}}$ represents main-pose cameras of all main-pose images from COLMAP. We then apply the resulting transformation to all Fast3R-predicted mixed-pose cameras:

$$\hat{P}_{\text{mix}} = (T^*, s^*) \cdot P_{\text{mix}} \tag{3}$$

This produces a globally aligned set of mixed-pose cameras in the main-pose COLMAP coordinate frame.

**Stage 2: Align the auxiliary-pose coordinate frame with the transformed mixed-pose coordinate frame (Figure 3(right))** Next, we align the COLMAP-estimated auxiliary cameras with their

transformed Fast3R counterparts. We solve

$$(T^*, s^*) = \mathcal{F}(P_{\text{aux}}, \hat{P}_{\text{mix}}^{\text{aux}}, P_{\text{aux}}), \tag{4}$$

where $P_{\text{aux}}$ represents auxiliary-pose cameras from COLMAP and $\hat{P}_{\text{mix}}^{\text{aux}}$ represents transformed auxiliary-pose cameras from Fast3R (after Stage 1) Applying this transformation gives the final aligned auxiliary camera poses:

$$\hat{P}_{\text{aux}} = (T^*, s^*) \cdot P_{\text{aux}} \tag{5}$$

At the end of this two-stage procedure, all camera poses of main-poses and auxiliary-poses are coherently transformed into the main-pose COLMAP coordinate system, providing a robust and consistent initialization for the subsequent local registration step.

## 3.2 GRADIENT-BASED LOCAL REGISTRATION REFINEMENT

To further refine auxiliary-pose cameras alignment after global registration, we adopt a two-stage optimization with silhouette-guided followed by RGB-guided refinement. First, we align the auxiliary-pose cameras by minimizing a silhouette loss between SAM2 masks and 3DGS-rendered masks of the auxiliary-pose views, which provides robust geometric cues to correct residual misalignment. We optimize a single global similarity transform (i.e., rotation, translation, scale) for the auxiliary-pose camera group while keeping the 3DGS parameters fixed. Second, we fine-tune the same transform using a photometric objective on the rendered RGB images to achieve the alignment of the details without altering scene geometry.

## 3.3 3DGS MODEL COMPLETION

Lastly, we complete the 3DGS model by fine-tuning the previous 3DGS with the registered main-pose and auxiliary-pose views. However, we might have a large unbalanced image set between the current auxiliary-pose views and the previous fused views. This might lead to biased optimization and poor reconstruction quality from auxiliary viewpoints. Thus, we adopt a simple but effective balanced sampling strategy. In each training iteration, we randomly sample a subset of images from the previous fused views equal in number to the current auxiliary-pose images. The process is repeated until the total iteration count (i.e., 30k) is reached. The resulting 3DGS model exhibits high-quality reconstruction with consistent geometry and radiance across all viewpoints.

## 4 EXPERIMENTS

**Dataset.** To evaluate pose accuracy, we construct a synthetic dataset comprising five diverse objects, using models collected from the Internet and placed within a casual room scene. For each object, we render two distinct poses, and for each pose we uniformly sample 150 camera viewpoints over a viewing hemisphere to emulate in-the-wild capture conditions. In addition, to assess the effectiveness of our method in real-world settings, we collect a complementary dataset of five physical objects. Four of these objects are captured in a photography studio using an automatic turntable, each with two object poses, while one object is captured in an in-the-wild setting with three object poses. For the synthetic data, we evaluate PFGS by comparing camera registration error and the visual quality of the resulting 3DGS models against baseline methods. We perform registration on all 150 images, and for novel view synthesis we adopt a train-test split, using 131 images for training and 19 images for evaluation.

**Metrics.** Camera-registration error is reported as the angular difference between axes and the position error in units in the synthetic scene, while 3DGS quality is measured with PSNR, SSIM, and LPIPS. An ablation study further analyzes the contribution of each pipeline component and evaluates the robustness of PFGS as the number of mixed-pose images varies. All experiments are run on an NVIDIA GeForce RTX 4090 GPU, and the combined global and local registration stages require 22.6 minutes on average.

## 4.1 QUANTITATIVE AND QUALITATIVE EVALUATION

On our multi-pose dataset, we compare PFGS against three baselines: (i) VGGT applied to the union of main- and auxiliary-pose images after background removal; (ii) Fast3R applied to the same

Table 1: **Quantitative comparison of camera-registration errors on different objects in the synthetic dataset.** Angles, $\Delta\theta$, are in degrees and distances, $\Delta p$, in units in the synthetic scene.

| Method | Lego | | | | Ship | | | | Violin | | | |
|---|---|---|---|---|---|---|---|---|---|---|---|---|
| | $\Delta\theta_x\downarrow$ | $\Delta\theta_y\downarrow$ | $\Delta\theta_z\downarrow$ | $\Delta p\downarrow$ | $\Delta\theta_x\downarrow$ | $\Delta\theta_y\downarrow$ | $\Delta\theta_z\downarrow$ | $\Delta p\downarrow$ | $\Delta\theta_x\downarrow$ | $\Delta\theta_y\downarrow$ | $\Delta\theta_z\downarrow$ | $\Delta p\downarrow$ |
| VGGT | 85.4469 | 51.4718 | 75.3138 | 2.9420 | 91.9138 | 71.6092 | 85.6433 | 3.4935 | 86.9461 | 74.4318 | 93.2568 | 5.4549 |
| Fast3r | 79.4932 | 75.4109 | 74.9609 | 6.9244 | 53.3989 | 44.9043 | 64.5457 | 2.4444 | 39.1897 | 59.6869 | 57.5947 | 3.8663 |
| COLMAP | 0.2542 | 0.2725 | 0.2997 | 0.0219 | 41.7649 | 47.5155 | 47.1624 | 2.1549 | 0.2435 | 0.2695 | 0.2762 | 0.0215 |
| Ours | 0.0135 | 0.0176 | 0.0157 | 0.0011 | 0.0158 | 0.0207 | 0.0190 | 0.0015 | 0.0242 | 0.0351 | 0.0320 | 0.0026 |

| Method | House | | | | Cottage | | | |
|---|---|---|---|---|---|---|---|---|
| | $\Delta\theta_x\downarrow$ | $\Delta\theta_y\downarrow$ | $\Delta\theta_z\downarrow$ | $\Delta p\downarrow$ | $\Delta\theta_x\downarrow$ | $\Delta\theta_y\downarrow$ | $\Delta\theta_z\downarrow$ | $\Delta p\downarrow$ |
| VGGT | 97.2216 | 61.3596 | 88.3416 | 41.9638 | 77.7085 | 76.0321 | 87.2480 | 5.4729 |
| Fast3r | 49.8034 | 55.0978 | 47.5854 | 3.5795 | 78.2134 | 100.6664 | 83.7597 | 5.9660 |
| COLMAP | 21.2253 | 17.4291 | 24.1649 | 2.4664 | 0.1217 | 0.1008 | 0.1423 | 0.0095 |
| Ours | 0.0098 | 0.0089 | 0.0089 | 0.0017 | 0.0696 | 0.0804 | 0.0790 | 0.0058 |

Table 2: **Quantitative comparison of 3DGS using different registration methods.**

| Method | Lego | | | Ship | | | Violin | | | House | | | Cottage | | |
|---|---|---|---|---|---|---|---|---|---|---|---|---|---|---|---|
| | PSNR↑ | SSIM↑ | LPIPS↓ | PSNR↑ | SSIM↑ | LPIPS↓ | PSNR↑ | SSIM↑ | LPIPS↓ | PSNR↑ | SSIM↑ | LPIPS↓ | PSNR↑ | SSIM↑ | LPIPS↓ |
| GT | 35.813 | 0.991 | 0.008 | 30.349 | 0.969 | 0.023 | 37.379 | 0.993 | 0.006 | 37.742 | 0.991 | 0.010 | 32.281 | 0.966 | 0.029 |
| COLMAP | 35.263 | 0.990 | 0.009 | 22.120 | 0.923 | 0.101 | 36.072 | 0.992 | 0.007 | 27.610 | 0.955 | 0.069 | 32.210 | 0.967 | 0.028 |
| Ours | 35.704 | 0.991 | 0.009 | 30.168 | 0.969 | 0.022 | 37.375 | 0.993 | 0.006 | 37.669 | 0.991 | 0.010 | 32.374 | 0.967 | 0.028 |

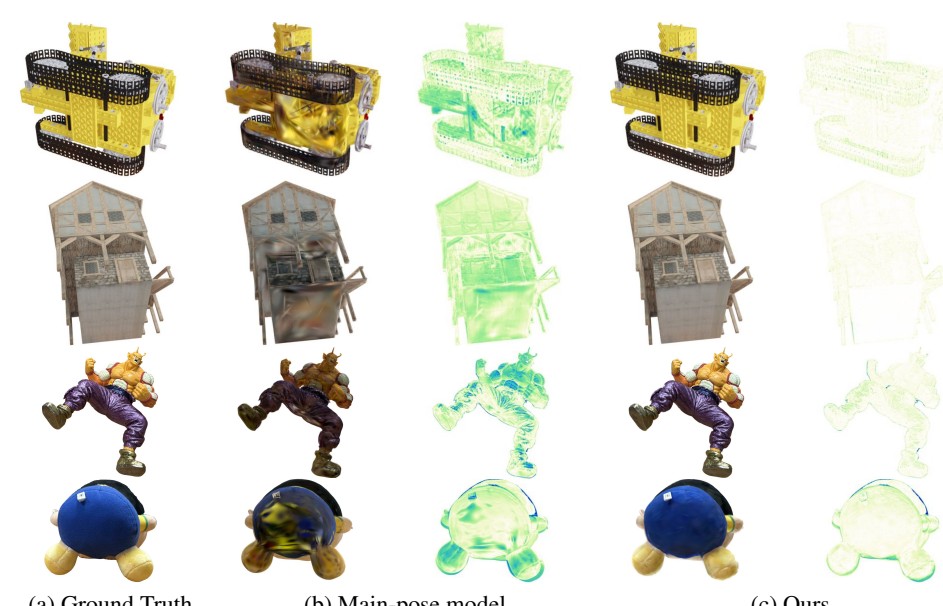

(a) Ground Truth  (b) Main-pose model  (c) Ours

Figure 4: **Visual quality of 3DGS reconstructions.** Panels show (a) ground-truth images, (b) the main-pose model, and (c) our method. Within each panel, the left image is a rendered view and the right image is a pixel-wise error map, where darker values denote larger $\ell_1$ differences from ground truth. Our reconstruction closely approaches ground truth fidelity while visibly surpassing the main-pose model.

union; and (iii) COLMAP run jointly on the main- and auxiliary-pose images using their foreground masks. Table 1 reports angular difference ($\Delta\theta_x, \Delta\theta_y, \Delta\theta_z$) and L2 position errors in meters ($\Delta p$) to the real-world benchmark for four representative objects. VGGT and Fast3R suffer from reduced feature coverage in masked images, yielding larger errors. In contrast, PFGS produces the lowest orientation and position errors across all objects. We then compare 3DGS reconstructions built with ground-truth (GT) camera poses, COLMAP estimates, and poses from PFGS in Table 2. PFGS attains PSNR, SSIM, and LPIPS close to the GT baseline, indicating accurate camera registration and consistent reconstruction quality. Figure 4 (top two rows) qualitatively compares a 3DGS model trained on main-pose captures only with PFGS trained on multi-pose captures. These results validate the effectiveness of PFGS. Additional evaluation are presented in Appendix A.1.

Table 3: **Ablation study of PFGS components**

| Method | $\Delta\theta_x \downarrow$ | $\Delta\theta_y \downarrow$ | $\Delta\theta_z \downarrow$ | $\Delta p \downarrow$ |
|---|---|---|---|---|
| w/o Mixed Pose Selection (Random) | 5.3513 | 7.7570 | 6.7388 | 0.6405 |
| w/o Local Refinement | 2.0455 | 2.4846 | 2.2947 | 0.1895 |
| Ours | 0.0266 | 0.0325 | 0.0309 | 0.0026 |

Table 4: **Visual quality of real-world multi-pose object reconstruction**

| Case | #Poses | Main-Pose 3DGS | | | Multi-Pose 3DGS | | |
|---|---|---|---|---|---|---|---|
| | | PSNR↑ | SSIM↑ | LPIPS ↓ | PSNR↑ | SSIM↑ | LPIPS ↓ |
| Marshall | 2 | 33.173 | 0.987 | 0.013 | 36.263 (+9.32%) | 0.992 (+0.48%) | 0.010 (-19.57%) |
| Hachiware | 2 | 29.678 | 0.951 | 0.058 | 31.170 (+5.03%) | 0.958 (+0.79%) | 0.055 (-6.27%) |
| Toy bricks | 2 | 31.374 | 0.980 | 0.015 | 32.590 (+3.88%) | 0.985 (+0.52%) | 0.013 (-10.85%) |
| Piccolo | 2 | 26.858 | 0.953 | 0.043 | 30.313 (+12.86%) | 0.971 (+1.95%) | 0.032 (-25.91%) |
| Shin-chan | 3 | 24.095 | 0.906 | 0.116 | 28.180 (+16.95%) | 0.926 (+2.16%) | 0.104 (-10.73%) |

## 4.2 ABLATION STUDY

We examine the impact of the mixed pose selection and local registration refinement as shown in Table 3. If we randomly select the mixed-pose images for the main-pose and auxiliary-pose for our fusion pipeline, it leads the accuracy of the predicted global registration to be off. Regarding to the local registration refinement, although our global registration can fuse the main-pose and auxiliary-pose, the two groups of camera poses still have a little misalignment. We demonstrate the local registration refinement improves the pose error from average angular error of around 2 degrees and position error around $0.18$ meters. However, the local registration refinement cannot fully correct errors introduced by random mixed-pose selection. Thus, we must use the global registration with the mixed-psoe image selection and local registration refinement to fully utilize the PFGS. In Appendix A.2, we also test PFGS with different sizes of the input image sets.

## 4.3 EVALUATION OF THE REAL-WORLD MULTI-POSE CAPTURES

We also examine the robustness of PFGS with real-world multi-pose captures. As mentioned above, we have four of the objects are captured in a photography studio using an automatic turntable with two object poses, while one object is captured in an in-the-wild setting with three object poses. For the mixed-pose selection, we set mixed-pose size $M$ and $N$ to 18 and the the top-ranked $K$ to 500. Figure 4 (bottom two rows) qualitatively compares a 3DGS model trained on main-pose captures only with PFGS trained on multi-pose captures, demonstrating that PFGS effectively completes the reconstructions with less difference to the ground truth.

## 5 CONCLUSION

We introduced Pose-Fused 3D Gaussian Splatting (PFGS), a pose-aware framework that achieves complete 3DGS object reconstructions from multi-pose image captures incrementally. At the core of PFGS is a global registration that fuses main-pose and auxiliary-pose with their mixed-pose images using a silhouette-consensus fusion. On synthetic datasets, PFGS surpasses baselines in camera pose recovery and 3DGS reconstruction quality. We also demonstrate high-quality 3DGS reconstructions from real-world multi-pose captures using PFGS.

Although PFGS achieves high-fidelity results, two limitations remain. First, the pipeline relies on COLMAP for initial camera poses; when feature matching fails in texture-poor scenes or with extremely sparse views, later registration stages can suffer. Second, we use Fast3R (Yang et al., 2025) to estimate mixed-pose cameras; when the chosen views lack sufficient viewpoint overlap or exhibit symmetric viewpoints, the estimated camera pose can become ambiguous or unstable. Future work will pursue an end-to-end design that handles all auxiliary poses in a single pass, replaces COLMAP with learning-based initialization, and incorporates lighter registration modules to reduce runtime while maintaining accuracy.

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

## A APPENDIX

### A.1 MORE QUALITATIVE EVALUATION

Additional qualitative results are provided in Figure 5, comparing synthetic (top three rows) and real-world (bottom three rows) data. We contrast a 3DGS model trained on main-pose captures with PFGS trained on multi-pose captures, further demonstrating that PFGS yields more complete reconstructions with smaller deviations from the ground truth.

### A.2 EVALUATION OF THE IMAGE SET SIZES

To assess the robustness of our proposed method under varying data availability, we conduct ablation studies on the size and balance of the input image sets. Specifically, we evaluate two scenarios: (i) balanced image sets, where the number of images per object pose is varied, and (ii) unbalanced image sets, where the auxiliary-pose image set are provided with fewer images than the main-pose image set.

**Balanced Image Sets.** We first investigate how performance changes as the total number of images per pose increases. In Table 5, the results show that our method maintains reliable performance across different set sizes. As expected, larger sets yield more accurate camera registration due to the availability of richer geometric information. This improvement in pose accuracy also translates into stronger novel view synthesis (NVS) results, with consistent gains in PSNR, SSIM, and LPIPS as the set size grows.

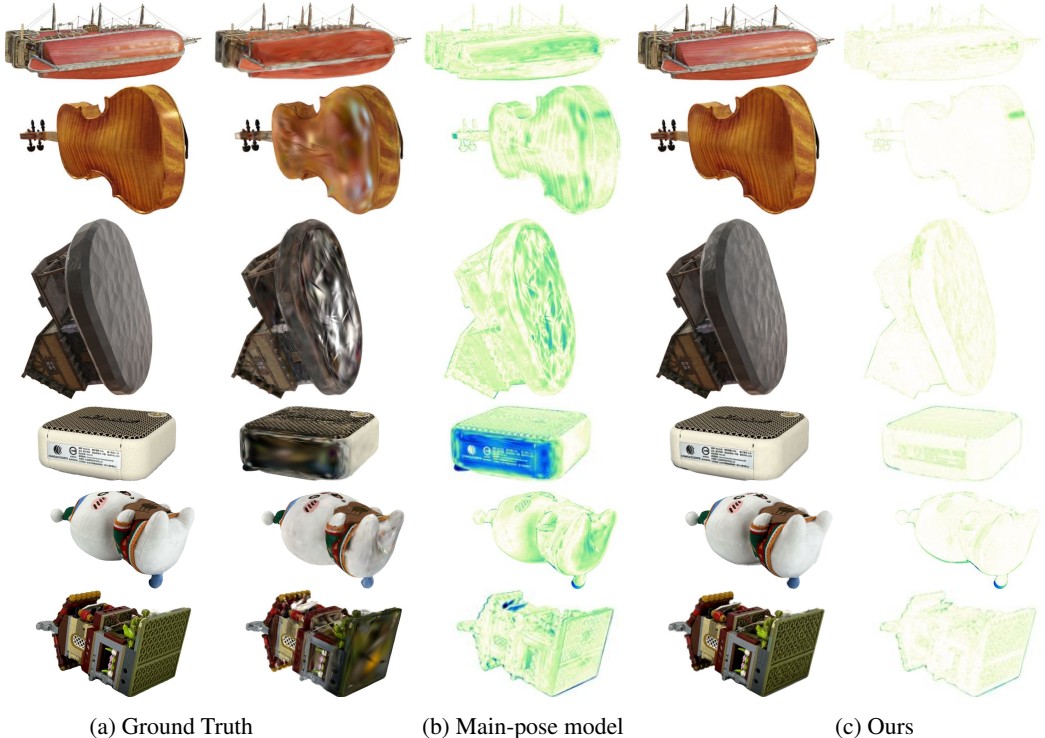

(a) Ground Truth         (b) Main-pose model         (c) Ours

Figure 5: **Visual quality of more 3DGS reconstructions.** Panels show (a) ground-truth images, (b) the main-pose model, and (c) our method. Within each panel, the left image is a rendered view and the right image is a pixel-wise error map, where darker values denote larger $\ell_1$ differences from ground truth. Our reconstruction closely approaches ground truth fidelity while visibly surpassing the main-pose model.

Table 5: **Robustness against different image set sizes**

| Set Size | Pose Error | | | | Novel View Synthesis | | |
|---|---|---|---|---|---|---|---|
| | $\Delta\theta_x \downarrow$ | $\Delta\theta_y \downarrow$ | $\Delta\theta_z \downarrow$ | $\Delta p \downarrow$ | PSNR↑ | SSIM↑ | LPIPS↓ |
| 60 | 0.2126 | 0.3074 | 0.2757 | 0.0162 | 32.884 | 0.976 | 0.019 |
| 90 | 0.0305 | 0.0349 | 0.0326 | 0.0030 | 34.144 | 0.982 | 0.016 |
| 150 | 0.0266 | 0.0325 | 0.0309 | 0.0026 | 34.658 | 0.982 | 0.015 |

Table 6: **Robustness against unbalanced image set sizes.** Each pair denotes (number of main views, number of auxiliary views).

| Set Size | Pose Error | | | | Novel View Synthesis | | |
|---|---|---|---|---|---|---|---|
| | $\Delta\theta_x \downarrow$ | $\Delta\theta_y \downarrow$ | $\Delta\theta_z \downarrow$ | $\Delta p \downarrow$ | PSNR↑ | SSIM↑ | LPIPS↓ |
| (150, 60) | 0.0197 | 0.0218 | 0.0231 | 0.0020 | 35.235 | 0.984 | 0.014 |
| (150, 90) | 0.0230 | 0.0262 | 0.0262 | 0.0023 | 35.008 | 0.983 | 0.015 |
| (150, 150) | 0.0266 | 0.0325 | 0.0309 | 0.0026 | 34.658 | 0.982 | 0.015 |

**Unbalanced Image Sets** Next, we examine the robustness of our approach when the main and auxiliary image set size are unbalanced. For this experiment, the auxiliary subsets are obtained by uniformly sampling images from the full set of 150. In Table 6, PFGS produces highly accurate camera poses and competitive novel view synthesis results across all configurations. Interestingly, while increasing the number of auxiliary images introduces slightly higher pose error, likely due to the added difficulty of aligning a larger number of viewpoints, the overall novel view synthesis

quality remains stable. We attribute the minor degradation in novel view synthesis performance to increased lighting inconsistencies when more auxiliary views are introduced.

These findings demonstrate that our method is robust to both the size and balance of input image sets. Even in challenging unbalanced scenarios, PFGS achieves consistently accurate pose estimation and strong novel view synthesis performance, highlighting its practical applicability under diverse data collection conditions.

## A.3 Effect of Mixed-Pose Set Size

Table 7: **Ablation study on mixed-pose image set size**

| Mixed-Pose Set Size | $\Delta\theta_x \downarrow$ | $\Delta\theta_y \downarrow$ | $\Delta\theta_z \downarrow$ | $\Delta p \downarrow$ |
|---:|---|---|---|---|
| 5 | 2.1805 | 3.0757 | 2.7001 | 0.1699 |
| 15 (Ours) | 0.0266 | 0.0325 | 0.0309 | 0.0026 |
| 30 | 0.9771 | 1.3663 | 1.1997 | 0.0770 |

We further ablate the size of the mixed-pose image set used during global registration (Table 7). This parameter controls the number of representative views drawn from different poses to guide the alignment process.

When the mixed-pose set is too small (e.g., 5 images), the registration quality degrades significantly. In this case, the limited reference pool fails to suppress the noise in the foundation model predictions, resulting in large pose errors. Conversely, when the set size is too large (e.g., 30 images), we observe that registration again becomes less reliable. We attribute this to the increased diversity of viewpoints, which introduces inconsistencies that confuse the foundation model and lead to noisier pose estimates.

Our default choice of around 15 images achieves the best trade-off, delivering highly accurate poses. This suggests that a moderate number of mixed poses provides sufficient cross-pose constraints without overwhelming the foundation model.

## A.4 Incomplete Registrations in the COLMAP Baseline

Table 8: **Registration coverage of COLMAP on the synthetic dataset**

| Case | Training Views | | Testing Views | |
|---|---|---|---|---|
| | All | Registered | All | Registered |
| Lego | 262 | 257 (98.1%) | 38 | 37 (97.4%) |
| Ship | 262 | 258 (98.5%) | 38 | 38 (100%) |
| Violin | 262 | 259 (98.9%) | 38 | 38 (100%) |
| House | 262 | 262 (100%) | 38 | 38 (100%) |
| Cottage | 262 | 243 (92.7%) | 38 | 36 (94.7%) |

Although COLMAP(Schönberger & Frahm, 2016) remains the strongest baseline in our comparison, it was unable to successfully register a very small fraction of cameras in the synthetic dataset. Table 8 summarizes the statistics, showing that in nearly all cases more than 97% of cameras were registered. These few missing views are rare and therefore have negligible impact on the overall results. Accordingly, in Table 1 and Table 2, we exclude missing cameras when computing COLMAP's pose error and novel view synthesis results, respectively.

## A.5 Memory Constraints of 3D Foundation Models

Feedforward 3D foundation models represent a significant advance and a paradigm shift for structure-from-motion (SfM). However, their applicability remains constrained by memory limitations, which restrict the number of input images. For example, while Fast3R(Yang et al., 2025) reports the ability to process large-scale datasets (e.g., up to 1,000 images) compared to earlier efforts such as Dust3R (Wang et al., 2024), this requires industry-grade GPUs. On consumer-grade

hardware, such as the RTX 4090, only a few hundred images can be accommodated (approximately 150 in our tests). Consequently, in the evaluation of camera pose registration reported in Table 1, VGGT(Wang et al., 2025) was run with 25 images per object pose, while Fast3R was run with 75 images per pose, all uniformly sampled from the complete set of 150 images per pose.

## A.6 DECLARATION OF LLM USAGE

We used large language models (LLMs) solely for polishing the text in this paper. Specifically, LLMs assisted in refining grammar, improving readability, and adjusting tone for academic writing. All research ideas, methodology, analyses, results, and terminology definitions presented in this work are original contributions from the authors and were not generated by LLMs.

