# OpenReview forum: "PFGS: Pose-Fused 3D Gaussian Splatting for Complete Multi-Pose Object Reconstruction"
_ICLR.cc/2026/Conference — ICLR 2026 Conference Withdrawn Submission_

### Official Review · Reviewer_Df9k · 2025-10-29

**Soundness:** 3
**Presentation:** 3
**Contribution:** 3
**Rating:** 4
**Confidence:** 4

**Summary:**

This paper presents PFGS (Pose-Fused 3D Gaussian Splatting), a pose-aware framework for reconstructing complete 3D objects from multi-pose image captures. Unlike conventional 3DGS approaches that assume a static object pose, PFGS handles scenarios where an object must be reoriented to reveal occluded regions.

The method iteratively fuses each auxiliary-pose sequence into a unified 3DGS representation of a main pose through three stages:

1. Global registration via a mixed-pose subset processed by foundation models (Fast3R, VGGT) and refined through a silhouette-consensus alignment.
2. Gradient-based local registration using silhouette and RGB consistency.
3. 3DGS completion through balanced sampling and fine-tuning.

Quantitative results on synthetic and real datasets demonstrate substantial improvements in pose accuracy (orders of magnitude lower error than VGGT/Fast3R) and enhanced 3D reconstruction.

**Strengths:**

- Proposing problem “Multi-pose reconstruction” seems challenging and interesting research direction.
- Novel framework to unify the poses of multiple sets of frames with efficient usage of foundation models
- Evaluations on well-curated synthetic and real datasets followed by ablation studies on each components.

**Weaknesses:**

1. **Illumination inconsistency ignored:** When an object’s pose changes, surrounding illumination directions and shading patterns also change. Fusing such data into a single 3DGS inevitably introduces appearance inconsistency. However, the method focuses on “photorealistic, view-consistent” results without modeling this radiometric gap.
2. **Insufficient analysis of baseline performance drop:** Table 1 shows VGGT and Fast3R failing dramatically on masked objects, but the authors do not analyze why (e.g., loss of background context or inconsistent normalization). A discussion on this could provide insight into the necessity of the fusion method.
3. **Limited scalability evidence:** While synthetic and real datasets are informative, all objects are relatively small-scale with moderate pose changes. It remains unclear whether PFGS generalizes to complex objects (e.g., articulated or deformable subjects) with extreme pose changes.

**Questions:**

1. How does the varying illumination by the pose change the impact the reconstruction result iof PFGS?
2. Why do VGGT and Fast3R perform much worse on masked (foreground-only) images?
3. How does PFGS handle cases where auxiliary poses differ by extreme rotations (e.g., >120°)?
4. Runtime comparison: How does the 22.6 min average compare to pure COLMAP or foundation-model-only approaches under the same GPU setup?

---

### Official Review · Reviewer_2Z7k · 2025-11-01

**Soundness:** 2
**Presentation:** 3
**Contribution:** 2
**Rating:** 4
**Confidence:** 4

**Summary:**

This paper introduces PFGS to solve the problem of 3D reconstructions with multiple-object-pose images. Given the main-pose and auxiliary-pose images, the authors propose a global registration stage that first find the visually overlapping views across poses, estimate the camera pose of mixed-pose images, then align auxiliary-pose views to main-pose coordinate system using silhouette-consensus fusion strategy. Gradient-based local registration optimization refines the camera poses, and a complete 3DGS model is trained with all poses.

**Strengths:**

1. The paper introduces a setting where the center object is at different poses to solve the occlusion and limited physical access issues comparing to a single-pose setting.
2. PFGS achieves better registration accuracy compared to VGGT, Fast3r, and COLMAP.
3. PFGS achieves better novel view synthesis compared to COLMAP-registered results and main-pose results.

**Weaknesses:**

1. The main technical contribution is the pose alignment. Similar topics discussed in Instant-Splat[1] and SPARS3R[2] regarding the train-test pose alignment.
2. A qualititative comparison with COLMAP-registered results would help understand the effectiveness of the proposed approach, as the results in Table 2 across different objects.

[1] Fan, Z., Cong, W., Wen, K., Wang, K., Zhang, J., Ding, X., ... & Wang, Y. (2024). Instantsplat: Sparse-view gaussian splatting in seconds. arXiv preprint arXiv:2403.20309.
[2] Tang, Y., Guo, Y., Li, D., & Peng, C. (2025). SPARS3R: Semantic Prior Alignment and Regularization for Sparse 3D Reconstruction. In Proceedings of the Computer Vision and Pattern Recognition Conference (pp. 26810-26821).

**Questions:**

1. Is there a need of registration of visually overlapping views across poses and then align instead of directly register the overlapping auxillary views to main-pose? If so, will using COLMAP for the registration of visually overlapping views lead to more accurate results, as the registration error from VGGT or Fast3r seem large in Table 1.
2. I don't entire understand the 3DGS-rendered the mask and SAM2 mask and how they contribute to the alignment accuracy. In figure 3 caption, it says "the lowest average IoU is selected".
3. Are the position units in synthetic dataset meters?
4. minor typos such as line 208 "we alignment". the draft merits a carefull proof-reading.

---

### Official Review · Reviewer_3Dzw · 2025-11-01

**Soundness:** 3
**Presentation:** 3
**Contribution:** 2
**Rating:** 4
**Confidence:** 4

**Summary:**

This paper proposes PFGS, a framework for reconstructing 3D objects from multi-pose image sets consisting of a main set and an auxiliary set. First, representative M and N images are selected from the main and auxiliary sets, respectively, using DINO feature similarity and geometric verification. These selected images form the initial mixed-pose set and are fed into Fast3R to estimate their camera poses. The poses of the mixed-pose images are aligned to the COLMAP poses of the main set by checking silhouette IoU. Similarly, the COLMAP poses for the auxiliary set are aligned to those of the transformed mixed-pose set via a similarity transform, followed by photometric refinement. Finally, the 3D Gaussians are further optimized using all images for improved reconstruction quality.

**Strengths:**

1. The task of reconstructing 3D objects from multi-pose image sets appears novel, as the background is no longer a reliable cue for registering images across different sets.
2. The proposed method is straightforward, and the experiments demonstrate its effectiveness both quantitatively and qualitatively.
3. The ablation studies confirm the contribution from each component.

**Weaknesses:**

1. The current method operates in a relatively easy setting:
(1) Both the main and auxiliary sets contain abundant images (e.g., more than 100) for COLMAP to function effectively. When the number of available images decreases, COLMAP may produce noisy poses or even fail entirely. It seems that the current system cannot handle such pose estimation failures. The reviewer is also curious about an ablation study where the number of images is reduced (e.g., fewer than 20). In such cases, learning-based approaches might perform better.
(2) The current formulation assumes static rigid objects and does not handle cases where the object exhibits articulation or non-rigid deformation across image sets.
2. The authors may consider discussing 3DGS-based object reconstruction with generative priors [1, 2], which can address challenges in sparse-view settings. Moreover, Zhao et al. [2] assume noisy camera poses as input and are able to handle outliers.

References:

[1] Tang et al., DreamGaussian: Generative Gaussian Splatting for Efficient 3D Content Creation, ICLR 2024.

[2] Zhao et al., Sparse-view Pose Estimation and Reconstruction via Analysis by Generative Synthesis, NeurIPS 2024.

**Questions:**

1. In the final stage (Sec. 3.3), are the camera poses optimized for all views to further improve alignment?

---

### Official Review · Reviewer_bk66 · 2025-11-01

**Soundness:** 2
**Presentation:** 2
**Contribution:** 1
**Rating:** 2
**Confidence:** 4

**Summary:**

This paper tackles the problem of reconstructing a 3D object from images captured under multiple object poses by introducing PFGS, a pipeline that fuses multi-pose camera tracks into a single 3DGS model. The system combines COLMAP initialization, DINO-based view similarity filtering, Fast3R for multi-view pose estimation, and a silhouette-guided similarity transform followed by RGB refinement to align poses across different capture sessions. Once aligned, 3DGS is fine-tuned to produce a unified representation. Experiments on a synthetic dataset and five real objects show improved completeness and pose accuracy relative to COLMAP, VGGT, and Fast3R. However, the method is heavily procedural, combines well-known building blocks, and provides only modest improvements over a strong COLMAP baseline.

**Strengths:**

- **Motivation and relevant problem setting.** - The paper addresses a realistic case where multi-pose captures are needed due to occlusions or physical constraints, which is a practical gap in existing 3DGS pipelines. The problem formulation of aligning auxiliary-pose views to a main-pose reconstruction is clear and grounded in real capture workflows.

- **Reasonable pipeline design with careful alignment of modules.** - The use of mixed-pose selection, global Fast3R alignment, and silhouette-based similarity transform makes the pipeline robust when naive COLMAP fails across poses. Ablations support that mixed-pose selection and local refinement meaningfully improve alignment quality.

**Weaknesses:**

- **Method is largely an engineering assembly of known components** - The pipeline strings together COLMAP, DINOv2, Fast3R, SAM2 masks, and 3DGS - each doing exactly what they were designed for. The silhouette-consensus fusion step is essentially a pose-space similarity alignment driven by mask IoU, which is useful but not conceptually novel.
- **Extremely high complexity and runtime.** The global and local registration pipeline averages 22.6 minutes on a 4090 (for ~300 images), which is expensive given that COLMAP can already handle many cases. The method delivers incremental improvements at a substantially higher computational cost, raising concerns about practicality.
- **Heavy reliance on accurate segmentation masks.**  Silhouette-guided registration/least-squares hinges on SAM2-style masks, leakage, thin structures, or reflectance will degrade both global alignment and local refinement. The paper does not analyze failure modes tied to mask quality.
- **Results are not strongly convincing relative to strong baselines.** - In several cases, COLMAP already provides near-GT camera accuracy, and improvements in NVS metrics are small (often ~0.3–0.8 dB PSNR).
- **Narrow and small-scale evaluation.** The core synthetic set covers five objects, and the real data also spans five objects (mostly controlled/turntable captures), which is too small to substantiate robustness claims. Improvements over baselines in NVS are modest and not statistically analyzed.
- COLMAP is used for initial main/auxiliary camera poses and also as a baseline; the pipeline explicitly relies on COLMAP and Fast3R, undermining the narrative that PFGS is a principled replacement rather than a wrapper.
- With the pace of 3D foundation models rapidly increasing (e.g., MAST3R/Spann3R/Fast3R improvements), the engineering-heavy nature of this pipeline risks becoming obsolete quickly. The paper positions itself as incremental bridge work, not actual progress.

**Questions:**

N/A

---

### Note · Authors · 2025-11-13

**Comment:**

We sincerely thank all the reviewers for their thoughtful comments, constructive suggestions, and the time they invested in evaluating our submission. We deeply appreciate the insights provided, many of which align with limitations we are already aware of and actively working to address. After careful consideration, we have concluded that we will not be able to adequately resolve all identified weaknesses within the rebuttal period. Therefore, we have decided to withdraw the paper from ICLR and continue refining it for submission to the next venue.

**Withdrawal Confirmation:**

I have read and agree with the venue's withdrawal policy on behalf of myself and my co-authors.